# Quasi-Orthogonal Model Merging for Continual Learning

## Abstract

Continual learning (CL) seeks to enable models to acquire new tasks sequentially without overwriting prior knowledge. Recently, model merging has emerged as a promising paradigm, where task vector, *i.e.*, parameter updates induced by fine-tuning, are combined across tasks. However, naive sequential merging often suffers from interference when task vectors overlap in conflicting directions. We introduce Quasi-Orthogonal Model Merging (QOMM), a unified framework that mitigates such interference through two complementary strategies. First, QOMM employs Singular Value Decomposition (SVD) to extract the dominant subspace of previously merged task vectors, and projects each new vector onto its approximate orthogonal complement. This Quasi-Orthogonal Projection (QOP) filters out conflicting directions, reducing interference. Second, QOMM integrates Attention-Exclusive Fine-Tuning (AEFT), which restricts updates to Transformer attention layers. This yields task vectors that are naturally more orthogonal, enhancing the effectiveness of QOP. By combining orthogonality-aware merging with attention-exclusive fine-tuning, QOMM achieves a better balance between stability (retaining past knowledge) and plasticity (adapting to new tasks). Experiments on standard CL benchmarks demonstrate that QOMM consistently outperforms prior methods. Our code will be released.

## 1 Introduction

Continual learning (CL), also known as lifelong learning, equips intelligent systems to operate in dynamic, non-stationary environments (De Lange et al., 2021; Masana et al., 2022; Van de Ven et al., 2022). Unlike conventional paradigms that assume access to a fixed training set, CL requires models to learn a sequence of tasks without revisiting data from earlier tasks. The central challenge is the *stability-plasticity dilemma* (Kim & Han, 2023), in which models must remain stable enough to retain prior knowledge while remaining plastic enough to assimilate new concepts.

Classical CL research has explored five broad families of strategies (Wang et al., 2024b): (1) regularization-based methods that constrain updates relative to the old model; (2) replay-based methods that approximate past data distributions; (3) optimization-based methods that shape training dynamics to balance tasks; (4) representation-based methods that learn robust, transferable features; and (5) architecture-based methods that adapt the network topology to accommodate new tasks while retaining prior knowledge. While effective to varying degrees, these approaches often rely on stored exemplars, auxiliary objectives, or task-specific modules that can complicate deployment.

Recently, model merging has emerged as a compelling alternative for CL (Liu & Soatto, 2023; Marczak et al., 2024; Marouf et al., 2024; Kleiman et al., 2025). Instead of continually updating a single network, a pre-trained base model is fine-tuned sequentially on each new task, and the resulting task-specific parameter updates (*i.e.*, "task vectors") are later integrated into a consolidated model. This paradigm avoids gradient interference during sequential training and shifts the stability–plasticity challenge to a post-hoc consolidation step. Even simple averaging merging strategy (Wortsman et al., 2022) has shown surprising robustness to forgetting in certain settings, and more sophisticated schemes such as MagMax (Marczak et al., 2024) merge task vectors by preserving maximum-magnitude updates to reduce forgetting.

Despite these advances, merging-based methods face a critical limitation—task interference during consolidation. When task vectors overlap in conflicting directions, naive merging can overwrite

or negate essential knowledge from earlier tasks, leading to performance degradation that worsens as the number of tasks increases. Heuristic rules (*e.g.*, magnitude-based selection) lack an explicit mechanism to guarantee compatibility between new and accumulated updates, leaving models vulnerable to negative transfer and effectively reintroducing the stability–plasticity trade-off at the merging stage itself.

In this work, we propose Quasi-Orthogonal Model Merging (QOMM), a unified framework that mitigates interference through two complementary mechanisms tailored to the structure of modern Transformers. First, QOMM performs an **orthogonality-aware merge** via Quasi-Orthogonal Projection (QOP). Given previously merged task vectors, we compute their dominant singular subspace using SVD and project each incoming task vector onto the approximate orthogonal complement of this subspace. This suppresses conflicting directions while preserving compatible information from the new task. The projection is *quasi*-orthogonal because it excludes only the dominant shared subspace. It can be interpreted as a low-rank approximation of an orthogonal projection, enabling a gradual and controllable balance between knowledge from new and previous tasks. Second, QOMM incorporates an Attention-Exclusive Fine-Tuning (AEFT) protocol that restricts updates to Transformer attention layers (Vaswani et al., 2017). By confining adaptation to attention (*e.g.*, query/key/value projections, and output projections), AEFT encourages task vectors that are naturally more disentangled and closer to orthogonal, thereby enhancing the effectiveness of the subsequent projection step. Together, these components enable QOMM to improve the stability-plasticity balance in merging-based CL: stability is reinforced by suppressing conflicting directions, and plasticity is preserved by retaining novel orthogonal components.

In summary, our contributions are summarized as follows:

- We introduce Quasi-Orthogonal Model Merging (QOMM), a continual learning (CL) framework based on model merging that tackles the central challenge of task interference during consolidation. QOMM achieves this through an orthogonality-aware merging strategy. To the best of our knowledge, it is the first CL framework of its kind to incorporate orthogonality-aware merging to effectively reduce task interference.

- At the core of QOMM, we introduce Quasi-Orthogonal Projection (QOP)-an orthogonality-aware merging mechanism in which the orthogonal projection is approximated in a low-rank manner, with controllable fidelity of approximation. This allows for a gradual and flexible balance between knowledge from new and previously learned tasks.

- To further enhance QOMM, we propose Attention-Exclusive Fine-Tuning (AEFT), which restricts updates to Transformer attention layers, yielding task vectors that are naturally more disentangled and closer to orthogonal, thereby amplifying the effectiveness of QOP.

Extensive experiments on standard CL benchmarks, including Split-CIFAR100, Split-ImageNetR, Split-CUB200, and Split-Cars, demonstrate that QOMM consistently outperforms both strong baselines and recent merging methods. On average, QOMM achieves 77.49% task-agnostic accuracy after the final task, surpassing the prior state-of-the-art approach by +2.99%. Ablation studies further verify that both components (*i.e.*, QOP and AEFT) are essential and synergistically contribute to the observed improvements.

## 2 RELATED WORK

**Model merging** refers to the process of consolidating multiple models which are typically fine-tuned from a common pre-trained initialization into a single network by integrating their parameters or task vectors. This paradigm avoids retraining from scratch and provides a scalable way to share knowledge across tasks. It has been proven effective and scalable in various domains, including language (Zhou et al., 2024), vision (Ye et al., 2023; Huang et al., 2023), and multimodal modeling (Yang et al., 2024; Chen et al., 2024). Early work perform simple parameter integration, such as element-wise parameter averaging (Wortsman et al., 2022), Fisher-weighted fusion (Matena & Raffel, 2022), predictive-divergence minimization (Jin et al., 2022), or arithmetic operations on task vectors (Ilharco et al., 2022a). While these methods laid the foundation of model merging, they generally lacked mechanisms to explicitly resolve conflicts between task updates, and thus remain vulnerable to task interference. Subsequent methods introduced heuristics to alleviate such conflicts. Ties-Merging prunes redundant parameters and resolves sign inconsistencies (Yadav et al.,

2023). DARE randomly drops and rescales parameters to reduce fusion conflicts (Yu et al., 2024). Consensus Merging filters unstable or harmful weights to improve robustness (Wang et al., 2024a). Although effective in some settings, these approaches remain heuristic and do not guarantee compatibility across task vectors. Interference among task vectors remains an open challenge.

**Model Merging for Continual Learning (CL).** Model merging has recently been adapted to CL as a post-hoc consolidation strategy that avoids task interference during training. Several methods achieve CL through distinct merging designs. TMC (Liu & Soatto, 2023) leverages linearly fine-tuned models (*i.e.*, tangent vectors) around a pre-trained initialization for continual learning. CoFiMA (Marouf et al., 2024) ensembles parameters across tasks using Fisher information. SFA (Kleiman et al., 2025) periodically merges models with earlier checkpoints during training. MagMax (Marczak et al., 2024) sequentially fine-tunes and merges parameters by maximum-magnitude selection. Despite their successes, existing merging-based CL methods remain vulnerable to task interference at the consolidation stage, where conflicting task vectors can inadvertently degrade previously acquired knowledge. Our work follows this "fine-tuning-then-merge paradigm" and proposes Quasi-Orthogonal Projection (QOP), an orthogonality-aware merging strategy that mitigates task interference by suppressing conflicting directions while preserving compatible information when merging new task vectors.

**Orthogonality in CL.** Orthogonality has long been recognized as a powerful principle for mitigating forgetting by reducing conflicts among task updates. OWM (Zeng et al., 2019) restricts weight updates to be orthogonal to the subspace spanned by past inputs. OGD (Farajtabar et al., 2020) maintains a subspace of past-task gradients and projects new gradients onto its orthogonal complement. SGP (Saha & Roy, 2023) combines orthogonal projections with scaled steps along important historical directions to enhance generalization. O-LoRA (Wang et al., 2023) learns low-rank subspaces that are explicitly orthogonal across tasks for parameter-efficient adaptation. Although effective, these methods operate at the level of gradients or parameter updates during training. In contrast, our approach applies orthogonality at the task vector level during merging, enabling explicit control over compatibility in post-hoc consolidation. Recent studies suggest that orthogonal task vectors can substantially improve merge quality (Xiong et al., 2024), and that restricting adaptation to Transformer attention modules enhances merge fidelity (Jin et al., 2025). Inspired by these insights, we hypothesize and confirm empirically that attention-only fine-tuning implicitly promotes task-vector orthogonality (see §4.3). Thus, we explicitly incorporate Attention-Exclusive Fine-Tuning (AEFT) in our orthogonality-aware merging framework to further strengthen subspace separation and reduce interference in continual model merging.

## 3 PRELIMINARY

### 3.1 PROBLEM FORMULATION

In model-merging-based continual learning (MMCL), the goal is to learn from a sequence of tasks without revisiting previous training data. Let $\mathcal{D} = \{D_i\}_{i=1}^{N}$ denote a sequence of $N$ disjoint task datasets. A pretrained base model with parameters $\Theta_0$ is fine-tuned sequentially on each dataset $D_i$, producing a task-adapted model $\Theta_i$. The corresponding parameter update, referred to as the task vector, is defined as $\Delta\Theta_i = \Theta_i - \Theta_0$. The collection of all task vectors is denoted by $\mathcal{T} = \{\Delta\Theta_i\}_{i=1}^{N}$. At task $i$, the merged model is represented as $\hat{\Theta}_i$, with its merged task vector given by $\Delta\hat{\Theta}_i = \hat{\Theta}_i - \Theta_0$. The merged model is iteratively updated by incorporating the new task vector $\Delta\Theta_i$ while mitigating interference with knowledge from previously integrated tasks. The objective is to obtain a final merged model $\hat{\Theta}_N$ that achieves good performance across all $N$ tasks without access to any individual task dataset $D_i$.

### 3.2 MOTIVATION

Our approach to model-merging–based continual learning is guided by three key insights. First, prior work has shown that orthogonal optimization is highly effective in mitigating parameter and gradient conflicts in continual learning. This motivates our Quasi-Orthogonal Projection (QOP) strategy, which employs approximate orthogonal projection of task vectors as a principled mechanism for incremental knowledge integration. Second, because this projection is constrained to the complement of the dominant singular subspace, the extracted orthogonal component is only approx-

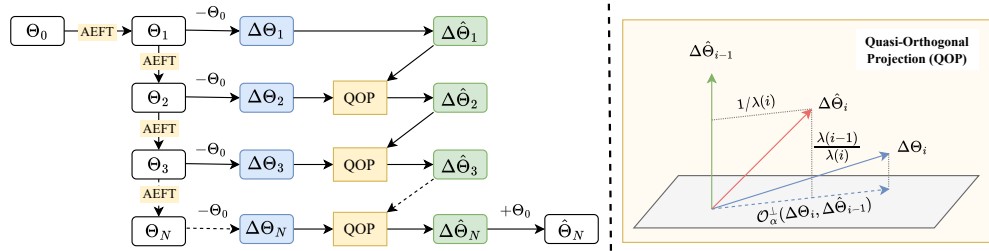

Figure 1: **Overview of Quasi-Orthogonal Model Merging (QOMM).** A pretrained model $\Theta_0$ is fine-tuned on sequential tasks with AEFT, producing task vectors $\Delta\Theta_i = \Theta_i - \Theta_0$. Each $\Delta\Theta_i$ is projected onto the approximate orthogonal complement of the dominant subspace of previous task vectors via $\mathcal{O}_\alpha^\perp$. The merged vector $\Delta\hat{\Theta}_i$ is then added to $\Theta_0$ to yield the final model $\hat{\Theta}_i$. See §4.

imate. Thus, the benefits of QOP can be further enhanced if task vectors are explicitly encouraged to become more orthogonal during fine-tuning. Third, Jin et al. (2025) demonstrate that restricting fine-tuning to Transformer attention modules substantially improves model-merging performance, while AWD (Xiong et al., 2024) shows that enforcing orthogonality among task vectors is critical for reducing interference and improving merge quality. We hypothesize that the gains reported in Jin et al. (2025) stem from an implicit increase in task vector orthogonality, a hypothesis strongly supported by our empirical findings. Building on these insights, we propose Quasi-Orthogonal Model Merging (QOMM), a framework that integrates QOP with Attention-Exclusive Fine-Tuning (AEFT) to address the stability–plasticity trade-off in continual learning through principled model merging.

# 4 METHODOLOGY

## 4.1 OVERVIEW

Our QOMM method comprises two complementary components. First, Quasi-Orthogonal Projection (QOP) extracts the dominant shared subspace among prior task vectors and suppresses conflicting directions during sequential merging. This ensures stable and effective knowledge integration. Second, Attention-Exclusive Fine-Tuning (AEFT) restricts adaptation to Transformer attention layers, encouraging the emergence of more orthogonal task vectors and thereby enhancing the effectiveness of QOP. Together, QOP and AEFT provide a computationally efficient strategy for merging task-specific updates, enabling continual learning systems to acquire new skills without sacrificing performance on previously learned tasks. The overall procedure is summarized in Algorithm 1 and illustrated in Figure 1.

## 4.2 QUASI-ORTHOGONAL PROJECTION

As shown in Figure 1, we fine-tune the model on a sequence of tasks and compute the corresponding task vectors $\Delta\Theta_i$ by subtracting the pre-trained model weights $\Theta_0$. These task vectors are then merged using Quasi-Orthogonal Projection (QOP), which relies on orthogonal projection. Finally, we apply the merged task vector to the pre-trained model to obtain the resulting model $\hat{\Theta}$. QOP addresses task interference at the merging stage by decomposing each new task vector into parallel and orthogonal components with respect to the subspace spanned by previous task vectors. Intuitively, the parallel component reflects weight changes aligned with prior task directions and is thus more likely to cause interference, while the orthogonal component points in a novel direction and is less likely to conflict with past knowledge. To formalize this, we perform a singular value decomposition (SVD) on the merged task vectors to identify their dominant subspace. The new task vector is then projected onto the orthogonal complement of this subspace, effectively filtering out directions that overlap with earlier tasks. By retaining only this orthogonal component (and discarding the conflicting parallel component), QOP achieves a better balance between stability (preserving prior knowledge) and plasticity (adapting to new tasks).

---

**Algorithm 1** QOMM : Quasi-Orthogonal Model Merging

---

1: $\Theta_1 = AEFT(\Theta_0)$
2: $\hat{\Theta}_1 = \Theta_1$
3: Scaling factor $\lambda_1 = 1$
4: **for** $i = 2$ to $N$ **do**
5:     $\Theta_i = AEFT(\Theta_{i-1})$
6:     Scaling factor $\lambda_i = \sqrt{i}$
7:     **for** Attention weight matrices $(W_0, \hat{W}_{i-1}, W_i) \in \mathbb{R}^{m \times n}$ in $(\Theta_0, \hat{\Theta}_{i-1}, \Theta_i)$ **do**
8:         $\Delta\hat{W}_{i-1} \leftarrow \hat{W}_{i-1} - W_0$
9:         $\Delta W_i \leftarrow W_i - W_0$
10:        $\Delta W_i^{\perp} \leftarrow \mathcal{O}^{\perp}\left(\Delta W_i, \Delta\hat{W}_{i-1}\right)$
11:        $\Delta\hat{W}_i \leftarrow \frac{\lambda_{i-1}\Delta\hat{W}_{i-1} + \Delta W_i^{\perp}}{\lambda_i}$
12:        $\hat{W}_{i-1} \leftarrow W_0 + \Delta\hat{W}_i$
13:     **end for**
14:     $\hat{\Theta}_i = \hat{\Theta}_{i-1}$
15: **end for**
16: **return** $\hat{\Theta}_N$

---

We conduct QOP at the layer level. At task $i$, for a given attention weight matrix $W \in \mathbb{R}^{m \times n}$ in layer $\ell$, we denote the pretrained parameters by $W_0^{(\ell)}$, the task-specific parameters (fine-tuned on dataset $D_i$) by $W_i^{(\ell)}$, and the cumulative merge after tasks $1{:}i-1$ by $\hat{W}_{i-1}^{(\ell)}$. The associated task vectors are always defined relative to the pretrained baseline: the task vector for task $i$ is $\Delta W_i^{(\ell)} = W_i^{(\ell)} - W_0^{(\ell)}$, and the cumulative task vector is $\Delta\hat{W}_{i-1}^{(\ell)} = \hat{W}_{i-1}^{(\ell)} - W_0^{(\ell)}$. The merged parameters after incorporating task $i$ are denoted by $\hat{W}_i^{(\ell)}$, with task vector $\Delta\hat{W}_i^{(\ell)}$. Unless otherwise noted, all matrix-level operations (*e.g.*, projections, scalings) are applied independently and identically to each attention matrix. For simplicity, we omit the superscript $(\ell)$ when the layer is clear from context.

Furthermore, We define $\mathcal{O}^{\perp}(\cdot, \cdot)$ as a projection operator that maps the task vector $\Delta Wi$ onto the orthogonal complement of the principal subspace spanned by the previously merged task vectors $\hat{W}_{i-1}$. In essence, $\mathcal{O}^{\perp}(\cdot, \cdot)$ isolates the orthogonal component of the current task vector relative to past updates, thereby enabling the integration of new knowledge while minimizing interference. To accomplish this, we begin by computing the full singular value decomposition (SVD) of the previously merged task vector:

$$\Delta\hat{W}_{i-1} = U_{i-1}\Sigma_{i-1}\left(V_{i-1}\right)^{\top}, \tag{1}$$

where $U_{i-1} \in \mathbb{R}^{m \times m}$ contains left singular vectors, $V_{i-1} \in \mathbb{R}^{n \times n}$ contains right singular vectors and $\Sigma_{i-1} \in \mathbb{R}^{m \times n}$ is a diagonal matrix of singular values in descending order.

We define $B_{pq} = u_p v_q^{\top}$, where $u_p$ and $v_q$ are the $p$-th and $q$-th columns of $U_{i-1}$ and $V_{i-1}$ respectively. $\{B_{pq}\}$ forms an orthonormal basis of the space of matrices with respect to the Frobenius inner product, and thus any matrix can be uniquely expressed as a linear combination of these basis elements. In particular,

$$\Delta\hat{W}_{i-1} = \sum_{p,q} c_{pq}B_{pq}, \quad c_{pq} = \langle\Delta\hat{W}_{i-1}, B_{pq}\rangle_F, \tag{2}$$

where $\langle\cdot, \cdot\rangle_F$ denotes the Frobenius inner product. The projection operator $\mathcal{O}^{\perp}(\cdot, \cdot)$ is constructed to preserve components orthogonal to the subspace spanned by the most significant singular directions:

$$\mathcal{O}^{\perp}\left(\Delta W_i, \Delta\hat{W}_{i-1}\right) = \Delta W_i - \sum_{j=1}^{r_\alpha}\langle\Delta W_i, B_{jj}\rangle_F \cdot B_{jj}, \tag{3}$$

where the threshold rank $r_\alpha$ is determined by:

$$r_\alpha = \min\left\{k \,\middle|\, \frac{\sum_{j=1}^{k}\sigma_j^2}{\sum_{j=1}^{\min(m,n)}\sigma_j^2} \geq \alpha\right\}, \quad \alpha \in [0, 1]. \tag{4}$$

In Eq. (4), $\alpha$ is the projection threshold hyper-parameter, which controls the balance between retaining existing knowledge and incorporating knowledge from new tasks.

The naive update rule of our QOP strategy is defined as

$$\Delta\hat{W}_i = \Delta\hat{W}_{i-1} + \mathcal{O}^{\perp}\left(\Delta Wi, \Delta\hat{W}_{i-1}\right), \qquad (5)$$

where the projection operator ensures that learning from new tasks predominantly occurs in the orthogonal complement of the dominant singular subspaces identified from previous tasks.

Although effective in promoting orthogonality, Eq. (5) suffers from an important limitation: as the number of tasks increases, the Frobenius norm of the merged update $\Delta\hat{W}_i$ grows monotonically. This accumulation leads to an undesirable drift, with the deviation of the merged model from the pre-trained initialization expanding unboundedly. To preserve a consistent magnitude of the merged model's parameter shift across tasks, it is necessary to regulate the Frobenius norm $||\Delta\hat{W}_i||_F$, ensuring stability throughout the merging process.

To address this, we propose scaling both the accumulated update $\Delta\hat{W}_{i-1}$ and the newly projected task update. Since each task's orthogonal contribution is treated as equally important, we adopt an adaptive normalization scheme, yielding the adaptive update rule:

$$\Delta\hat{W}_i = \frac{\lambda_{i-1}\Delta\hat{W}_{i-1} + \mathcal{O}^{\perp}\left(\Delta W_i, \Delta\hat{W}_i\right)}{\lambda_i}, \qquad (6)$$

where the scaling factor is defined recursively as $\lambda_1 = 1$ and $\lambda_i = \sqrt{i}$. This formulation ensures that the merged model remains close to the pre-trained model while progressively integrating task-specific knowledge in a balanced and controlled manner.

Finally, the merged model parameters after incorporating $i$ tasks are obtained as

$$\hat{W}_i = W_0 + \Delta\hat{W}_i. \qquad (7)$$

### 4.3 ATTENTION-EXCLUSIVE FINE-TUNING

While Quasi-Orthogonal Projection (QOP) provides a principled mechanism for reducing interference, its effectiveness is ultimately bounded by the quality of task-vector orthogonality. As shown in Eq. (3) and Eq. (4), the orthogonal projection only removes overlap with the dominant singular subspace, making the resulting updates approximate rather than exact. This motivates the need for a complementary strategy that can actively promote the emergence of orthogonal task vectors during fine-tuning. Attention-Exclusive Fine-Tuning (AEFT) addresses this need by restricting task-specific parameter updates to the Transformer's attention modules. This choice is motivated by two observations: (i) Constraining fine-tuning to Transformer attention modules substantially improves merged model performance (Jin et al., 2025), and (ii) Enforcing orthogonality among task vectors is critical for mitigating interference and enhancing merge quality (Xiong et al., 2024). We hypothesize that constraining fine-tuning to attention-related linear layers, rather

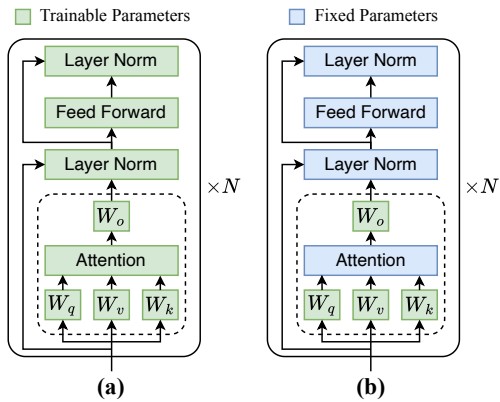

Figure 2: Two fine-tuning paradigms. (a) Full-Model Fine-Tuning (FMFT) where all the parameters will be updated. (b) Attention-Exclusive Fine-Tuning (AEFT) where only $W_q, W_k, W_v, W_o$ will be updated. See §4.3 for details.

than performing full model updates, yields more orthogonal task vectors and thus enhancing the performance of QOP.

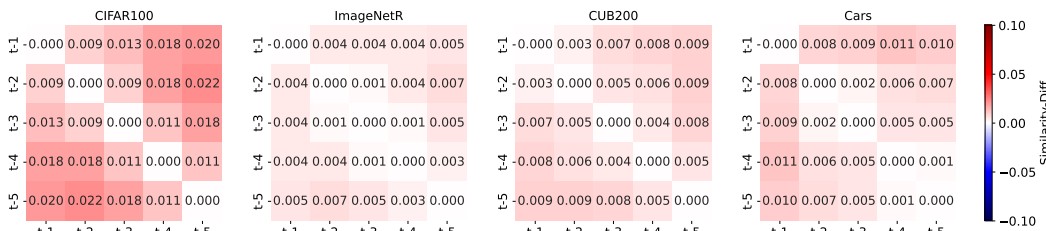

Figure 3: Cosine similarity difference matrices ($M^{\text{Diff}} = M^{\text{FMFT}} - M^{\text{AEFT}}$) across four benchmarks. Positive values (red) indicate that FMFT produces higher similarity between task vectors than AEFT, meaning lower orthogonality. This pattern confirms that AEFT encourages task vectors to be more orthogonal, thereby reducing interference. See §4.3 for details.

Two fine-tuning paradigms are illustrated in Figure 2. To evaluate our hypothesis, we compare Full-Model Fine-Tuning (FMFT) with Attention-Exclusive fine-tuning (AEFT) across four datasets (CIFAR-100, ImageNet-R, CUB-200, and Cars) in a continual learning setup. Each dataset is divided into five sequential tasks, and for each paradigm, we derive five task vectors. We then compute the cosine similarity matrices $M^{\text{FMFT}}, M^{\text{AEFT}} \in \mathbb{R}^{5 \times 5}$, where $M_{ij}^{\text{FMFT}} = \cos(\Delta W_i^{\text{FMFT}}, \Delta W_j^{\text{FMFT}})$ and $M_{ij}^{\text{AEFT}} = \cos(\Delta W_i^{\text{AEFT}}, \Delta W_j^{\text{AEFT}})$, with $\cos(\cdot, \cdot)$ denoting cosine similarity. Figure 3 presents the difference matrix $M^{\text{Diff}} = M^{\text{FMFT}} - M^{\text{AEFT}}$. The results indicate that task vectors from full fine-tuning exhibit higher similarity (*i.e.*, lower orthogonality) than those from attention-exclusive updates. This trend persists when scaling to 10/20/50 tasks (see §A.2), confirming that attention-constrained fine-tuning yields more orthogonal task vectors, thus supporting our hypothesis.

## 5 EXPERIMENTS

### 5.1 EXPERIMENTAL SETUPS

**Datasets.** To ensure consistency with prior work, we follow the experimental setup of Mag-Max (Marczak et al., 2024), adopting the same datasets and task configurations for both class-incremental learning (CIL) and domain-incremental learning (DIL) settings. For CIL, we use CIFAR100 (Krizhevsky et al., 2009) and ImageNet-R (Hendrycks et al., 2021) as generic image classification benchmarks, and CUB200 (Wah et al., 2011) and Cars (Krause et al., 2013) as fine-grained datasets. Each dataset is partitioned into $N$ disjoint subsets of classes, where $N \in \{5, 10, 20, 50\}$ for generic benchmarks and $N \in \{5, 10, 20\}$ for fine-grained benchmarks (due to their smaller size). For DIL, we adopt DomainNet (Peng et al., 2019) as the benchmark dataset and divide it into six tasks based on domains (clipart, infographics, painting, quickdraw, real, and sketch).

**Baselines.** We evaluate QOMM against well-established CL baselines, including LwF (Li & Hoiem, 2017) and EWC (Kirkpatrick et al., 2017), as well as recent model merging strategies such as ModelSoup (Wortsman et al., 2022), Task Arithmetic (TA) (Ilharco et al., 2022a), and TIES-Merging (TIES) (Yadav et al., 2023). In addition, we compare with MagMax (Marczak et al., 2024) and its two variants: RandMix, which randomly samples each parameter from one of the fine-tuned models, and MaxAbs, which applies independent fine-tuning instead of sequential adaptation. Finally, we report zero-shot performance, reflecting the capability of the pre-trained model, and joint performance, corresponding to a model fine-tuned on the entire dataset.

**Implementation details.** We use the CLIP pre-trained model (Radford et al., 2021) with a ViT/B-16 image encoder (Dosovitskiy et al., 2021). Following the fine-tuning procedure of Ilharco et al. (2022b), we adapt the image encoder using AdamW with weight decay and a cosine annealing learning rate schedule. For each split task of each dataset, we use the following training configurations: CIFAR-100 (batch size 64, learning rate $1.8 \times 10^{-5}$, 20 epochs, weight decay 0.09), ImageNet-R (batch size 64, learning rate $1.7 \times 10^{-5}$, 20 epochs, weight decay 0.09), CUB-200 and Cars (batch size 32, learning rate $3 \times 10^{-5}$, 24 epochs, weight decay 0.09), and DomainNet (batch size 64, learning rate $1.8 \times 10^{-5}$, 20 epochs, weight decay 0.09). We use the final classification layer output by CLIP's text encoder and keep it frozen during fine-tuning, following (Ilharco et al., 2022b). This

Table 1: Comparison on different methods. Our method outperforms other continual learning methods and merging-based approaches on a wide variety of class-incremental scenarios. We report task-agnostic accuracy (%) after the final task. Our results in **bold** are the best across all tasks, and the prior best performing results are underlined. See §5.2 for details.

| Method | CIFAR100 | | | | ImageNet-R | | | | CUB200 | | | Cars | | | Avg |
|---|---|---|---|---|---|---|---|---|---|---|---|---|---|---|---|
| | /5 | /10 | /20 | /50 | /5 | /10 | /20 | /50 | /5 | /10 | /20 | /5 | /10 | /20 | |
| Zero-shot | 66.91 | | | | 77.73 | | | | 56.08 | | | 64.71 | | | 67.21 |
| Joint | 90.94 | | | | 87.55 | | | | 81.57 | | | 88.21 | | | 87.38 |
| LwF | 83.25 | 74.35 | 72.05 | 68.84 | 81.15 | 82.97 | 81.82 | 80.32 | 65.12 | 60.67 | 58.90 | 71.72 | 69.84 | 62.98 | 72.36 |
| EWC | 84.41 | 76.24 | 75.39 | 72.97 | 82.15 | 82.42 | 81.48 | 81.47 | 59.10 | 54.49 | 53.31 | 69.46 | 60.78 | 57.42 | 70.79 |
| RandMix | 81.55 | 77.04 | 75.36 | 72.91 | 83.10 | 81.88 | 80.18 | 78.50 | 59.86 | 58.53 | 58.08 | 67.32 | 65.62 | 64.95 | 71.78 |
| MaxAbs | 81.95 | 76.75 | 74.39 | 73.04 | 83.03 | 82.33 | 80.92 | 79.33 | 60.15 | 58.01 | 56.59 | 67.36 | 63.55 | 58.95 | 71.17 |
| ModelSoup | 81.41 | 77.04 | 75.29 | 72.92 | 83.08 | 81.87 | 80.27 | 78.53 | 59.77 | 58.44 | 58.01 | 67.37 | 65.59 | 64.88 | 71.85 |
| TIES | 81.72 | 77.23 | 74.66 | 73.76 | 83.08 | 82.27 | 80.83 | 79.57 | 60.94 | 58.22 | 56.97 | 70.45 | 64.90 | 61.17 | 71.84 |
| MAGMAX | 84.16 | 80.41 | 78.49 | 76.75 | 83.60 | 83.33 | 82.27 | 81.75 | 63.89 | 60.74 | 58.90 | 73.61 | 69.28 | 65.84 | 74.50 |
| **QOMM (ours)** | **85.20** | **83.09** | **80.69** | **77.84** | **85.77** | **84.52** | **83.05** | **81.78** | **69.02** | **64.84** | **62.10** | **80.54** | **74.90** | **71.50** | **77.49** |
| **Performance Δ** | **+0.79** | **+2.68** | **+2.20** | **+1.09** | **+2.17** | **+1.19** | **+0.78** | **+0.03** | **+3.90** | **+4.10** | **+3.20** | **+6.93** | **+5.06** | **+5.66** | **+2.99** |

Table 2: DIL Performance (%) of different methods on DomainNet. See §5.2 for details.

| Dataset | LwF | EWC | RandMix | MaxAbs | Avg | TIES | MAGMAX | Ours |
|---|---|---|---|---|---|---|---|---|
| DomainNet (DIL) | 69.67 | **70.74** | 64.31 | 67.51 | 64.98 | 66.42 | 69.00 | 69.32 |

fine-tuning recipe preserves the open-vocabulary nature of the model and does not harm the accuracy compared to training the classification layer (Ilharco et al., 2022b). Each experiment is run on a single NVIDIA GeForce RTX 4090 GPU.

**Memory Complexity.** As shown in Figure 1, during the model merging-based continual learning process, only a fixed set of models must be maintained in memory at any step $i$: the current merged version, the incoming model for merging, and the original pre-trained base model. This strategy results in memory complexity of $\mathcal{O}(|\Theta|)$, with $|\Theta|$ denoting the parameter count of a single model. Notably, the memory footprint stays invariant to the total count of downstream tasks being processed.

## 5.2 Main Results

**Class-incremental learning (CIL).** Table 1 summarizes the task-agnostic accuracies (%) across four widely used class-incremental benchmarks. Our method consistently outperforms all continual learning and merging-based baselines, achieving the highest accuracy in every setting. Averaged over all datasets and task splits, our approach reaches 77.49%, a +2.99% gain over the second-best method (MAGMAX). On CIFAR100, our method yields the best results across all splits, with margins of up to +2.68% compared to the second-best approach, demonstrating strong scalability as the number of tasks increases. On the more challenging ImageNet-R benchmark, our approach again secures the top performance with consistent gains, highlighting robustness in large-scale recognition. For fine-grained datasets, improvements are even more pronounced: on CUB200, our method exceeds prior approaches by +3-4%, while on Cars it achieves the largest margins in the table, outperforming alternatives by +5-6%. These substantial gains indicate that our method is particularly effective in domains with subtle inter-class variations and high visual similarity. Overall, the results establish our approach as a new state of the art for class-incremental learning, providing consistent, robust, and significant improvements across both coarse- and fine-grained benchmarks.

**Domain-incremental learning (DIL).** Table 2 presents the results on DomainNet under the DIL setting. EWC achieves the highest performance with 70.74%, while our method delivers a competitive 69.32%. Notably, our approach surpasses strong merging-based methods such as MAGMAX (69.00%), TIES (66.42%), and MaxAbs (67.51%), as well as the simple averaging baseline (64.98%). These findings indicate that our method remains highly effective on challenging domain-incremental scenarios, performing on par with the strongest continual learning approaches.

## 5.3 Ablation Study

In this section, we evaluate the effectiveness of the two proposed components (*i.e.* QOP and AEFT) as well as the influence of the hyperparameter $\alpha$ introduced in §4. The experiments are conducted

Table 3: CIL Performance (%) of MagMax, QOP, and QOP + AEFT on CIFAR100 and CUB200 with different number of tasks. See §5.3 for details.

| Method | CIFAR100 | | | | CUB200 | | |
|---|---|---|---|---|---|---|---|
| | 5 | 10 | 20 | 50 | 5 | 10 | 20 |
| MagMax | 84.16 | 80.41 | 78.49 | 76.75 | 63.89 | 60.74 | 58.90 |
| QOP | 82.82 | 78.05 | 69.61 | 52.60 | 64.74 | 57.44 | 45.79 |
| QOP + AEFT | **85.20** | **83.09** | **80.69** | **77.84** | **69.02** | **64.84** | **62.10** |

on two representative datasets: the generic image classification benchmark CIFAR100 and the fine-grained recognition dataset CUB200.

**Analysis of QOP and AEFT.** Table 3 presents CIL results on CIFAR100 and CUB200 with varying task numbers. When using QOP alone, the performance is comparable to MagMax when the number of tasks is small, for example at 5 tasks on both datasets. However, as the number of tasks increases, QOP suffers from a severe drop in accuracy, whereas MagMax degrades more gradually. This phenomenon can be attributed to the fact that QOP is based on an approximate orthogonal projection, which becomes less effective as task interference grows. By contrast, incorporating AEFT into QOP substantially alleviates this issue. QOP + AEFT consistently outperforms both QOP and MagMax across all task numbers, demonstrating not only higher accuracy in the low-task regime but also much greater robustness as the number of tasks increases. This improvement arises because AEFT enforces stronger orthogonality among task vectors, thereby reducing interference and preserving performance even under a large number of tasks.

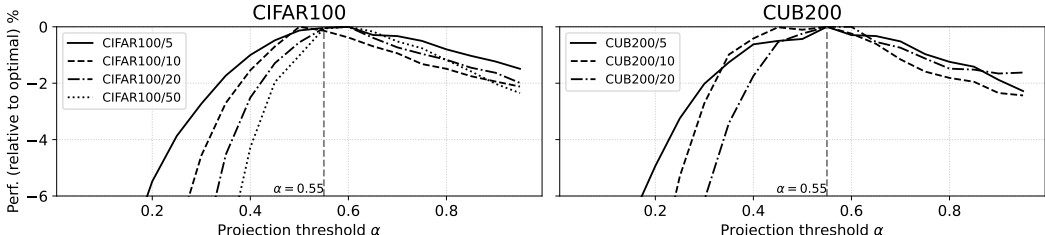

Figure 4: **Sensitivity to projection threshold $\alpha$.** Results are reported on CIFAR100 (left) and CUB200 (right) under different task splits. Accuracy is measured relative to the optimal $\alpha$ for each setting, and performance remains robust within $\alpha \in [0.5, 0.6]$, with $\alpha = 0.55$ yielding near-optimal accuracy in most cases. See §5.3 for details.

**Sensitivity to Projection Threshold $\alpha$.** We investigate the impact of the projection threshold $\alpha$ on QOMM's performance across CIFAR100 (generic classification) and CUB200 (fine-grained classification), as shown in Figure 4. The results demonstrate that QOMM's performance is relatively stable for $\alpha \in [0.5, 0.6]$, with $\alpha = 0.55$ consistently providing near-optimal accuracy across different task splits. Based on this observation, we fix $\alpha = 0.55$ for all experiments in our work.

## 6 CONCLUSION

We presented Quasi-Orthogonal Model Merging (QOMM), an orthogonality-aware merge method for continual learning that explicitly addresses task interference during consolidation. QOMM integrates two complementary components: Quasi-Orthogonal Projection (QOP), which suppresses conflicting directions by projecting new task vectors onto the approximate orthogonal complement of previously merged subspaces, and Attention-Exclusive Fine-Tuning (AEFT), which amplifies the effectiveness of QOP by restricting adaptation to Transformer attention layers to produce more orthogonal task vectors. Together, QOMM achieves a better balance between stability (retaining past knowledge) and plasticity (adapting to new tasks). Extensive experiments on standard CL benchmarks demonstrate that QOMM consistently outperforms existing methods, and ablations confirm the necessity of both QOP and AEFT. We hope that this work encourages further exploration of orthogonality-aware strategies for advancing model merging-based CL.

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

## A  APPENDIX

### A.1  LLM USAGE

We used LLM solely for polishing the writing of this paper, including improving grammar, clarity, and style. The model was not involved in research ideation, experimental design, analysis, or the generation of scientific results. All content and claims in the paper are the responsibility of the authors.

### A.2  SIMILARITY DIFF

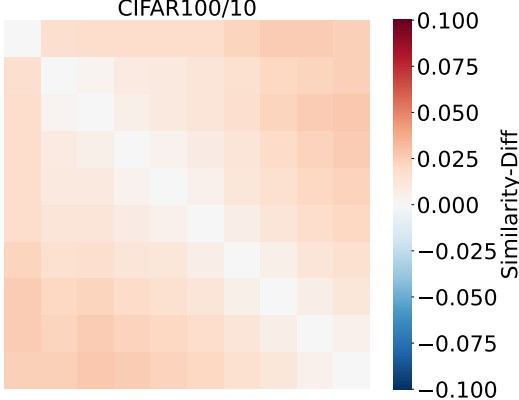

Figure 5: Cosine similarity difference matrix ($M^{\text{Diff}} = M^{\text{FMFT}} - M^{\text{AEFT}}$) for CIFAR100/10

Figures 5, 6 and 7 present cosine similarity difference matrices ($M^{\text{Diff}} = M^{\text{FMFT}} - M^{\text{AEFT}}$) for CIFAR100 split into 10, 20, and 50 tasks, respectively. In all cases, off-diagonal entries are predominantly red, indicating that FMFT task vectors are consistently more similar (less orthogonal) than those from AEFT, supporting hypothesis $H_1$.

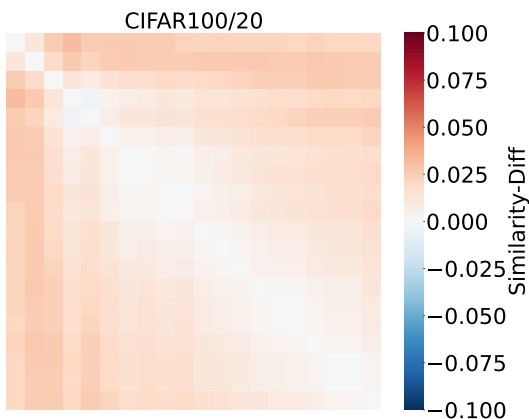

Figure 6: Cosine similarity difference matrix ($M^{\text{Diff}} = M^{\text{FMFT}} - M^{\text{AEFT}}$) for CIFAR100/20

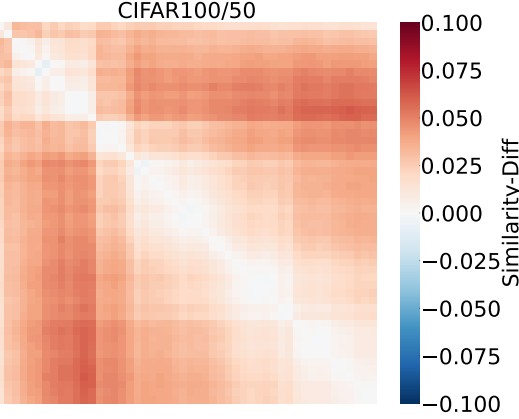

Figure 7: Cosine similarity difference matrix ($M^{\text{Diff}} = M^{\text{FMFT}} - M^{\text{AEFT}}$) for CIFAR100/50

