# OpenReview forum: "Quasi-Orthogonal Model Merging for Continual Learning"
_ICLR.cc/2026/Conference — ICLR 2026 Conference Withdrawn Submission_

### Official Review · Reviewer_MYPy · 2025-10-28

**Soundness:** 1
**Presentation:** 2
**Contribution:** 2
**Rating:** 2
**Confidence:** 4

**Summary:**

This paper proposes Quasi-Orthogonal Model Merging (QOMM) for merging-based continual learning. Specifically, Quasi-Orthogonal Projection (QOP) is introduced to mitigate conflicts when merging task vectors, and Attention-Exclusive Fine-Tuning (AEFT) is utilized to enhance the orthogonality among task vectors.

**Strengths:**

1. The motivation of the proposed method is clear. Parameter optimization in an orthogonal space is a well-recognized and effective approach in continual learning.
2. The paper is clearly written and easy to follow.

**Weaknesses:**

1. My major concern lies in the absence of evaluation metrics. In continual learning, it is important to assess the trade-off between stability and plasticity using both average accuracy and backward transfer (BWT). Neglecting the BWT metric undermines the credibility of the experimental results. Moreover, the experiments are not repeated with different seeds, nor are the mean and variance reported. As the randomness of dataset partition and model training can significantly affect the continual learning process, this omission raises concerns about the robustness of the proposed method's performance.
2. In the setting of domain-incremental learning (DIL), the proposed method does not achieve better performance compared to pervious works. Notably, EWC was introduced as early as 2017, yet the authors provide no analysis or discussion to explain why their method fails to surpass such earlier baselines.
3. Several existing works in continual learning adopt similar ideas of model merging (often referred to as interpolation, ensembling, etc.), but none of these are included in the experimental comparison. The authors are strongly encouraged to include a more comprehensive discussion on the relationship between QOMM and these prior works, as many relevant studies are currently omitted.

[1]  Stojanovski, Z., Roth, K., and Akata, Z. Momentum-based weight interpolation of strong zero-shot models for con tinual learning. In NeurIPS Workshop, 2022.

[2]  Simon, C., Faraki, M., Tsai, Y.-H., Yu, X., Schulter, S., Suh, Y., Harandi, M., and Chandraker, M. On generalizing beyond domains in cross-domain continual learning. In CVPR, 2022.

[3]  Lee, J., Joo, D., Hong, H. G., and Kim, J. Residual continual learning. In AAAI, 2020.

[4]  Lin, G., Chu, H., and Lai, H. Towards better plasticity stability trade-off in incremental learning: A simple linear connector. In CVPR, 2022.

[5]  Marouf, I. E., Roy, S., Tartaglione, E., and Lathuili`ere, S. Weighted ensemble models are strong continual learners. In ECCV, 2024.

[6] Li, M., Lu, Y., Dai, Q., Huang, S., Ding, Y., Lu, H., BECAME: Bayesian Continual Learning with Adaptive Model Merging. In ICML, 2025.

**Questions:**

1. In the ablation study and Table 3, a naive baseline, i.e., a variant without QOP, should be added to more clearly demonstrate the contribution of QOP to overall performance.
2. The model used in experiments is a pre-trained CLIP. Considering the datasets used in experiments are relatively simple for this powerful model, I am wondering if the proposed method can attain consistent performance when using different models, such as ViT model from scratch, or smaller models.

---

### Official Review · Reviewer_RVGH · 2025-11-01

**Soundness:** 2
**Presentation:** 2
**Contribution:** 2
**Rating:** 2
**Confidence:** 3

**Summary:**

The paper proposes a continual learning method based on model merging. To mitigate the interference caused by sequential merging, it proposes the Quasi-Orthogonal Model Merging method (QOMM) that projects the new task vectors to the basis of previously merged task vectors, and filters out components parallel to previous task vectors. To encourage orthogonality of task vectors, the paper restricts updates only to Transformer attention layers. The paper conducts experiments over regularization-based CL methods and model merging methods. Results show that the proposed method achieves better performance in CIL.

**Strengths:**

1. Applying model merging methods to CL is an emerging topic and has potential to reduce forgetting for large scale models.

2. The experiments are conducted over both CIL and DIL settings, with comparison to many existing models. The ablation study is included.

**Weaknesses:**

1. It is unclear why Quasi-Orthogonal projection (QOP) achieves a better balance between stability and plasticity.
- Filtering new task vector components that are in the same directions of previous task vectors is a strong regularization, which may harm the model’s plasticity. In addition, scaling the parameter updates to be close to the pre-trained model may intensify the problem.
- In experiments Table 3, using QOP only underperforms EWC when the number of data subsets is more than 20.

2. The hypothesis ‘constraining fine-tuning to attention-related linear layers yields more orthogonal task vectors’ is interesting. However, to support the hypothesis, some details of figure 3 are missing.
- What are the settings of FMFT and AEFT training? Do FMFT and AEFT share the same learning rates and training epochs? Since the cosine similarity ranges in [-1,1], should the $M^{diff}$ be computed over absolute $M^{FMFT}$ and $M^{AEFT}$?

3. The experiments only compare with regularization based CL baselines (LwF and EWC) besides model merging methods. It could be helpful to see how QOMM performs compared to more recent CL methods.

**Questions:**

1. In experimental result Table 3, using QOP only has a large performance gap compared to  QOP + AEFT.  Does that mean the improvement of QOMM only comes from AEFT? Will adding AEFT to other methods like EWC, MAGMAX achieve better performance than QOP + AEFT (QOMM)?

---

### Official Review · Reviewer_fCmW · 2025-11-03

**Soundness:** 2
**Presentation:** 3
**Contribution:** 1
**Rating:** 2
**Confidence:** 4

**Summary:**

The paper introduces Quasi-Orthogonal Model Merging (QOMM), a continual learning framework that aims to mitigate task interference when merging task-specific models. The approach operates in two stages: first, it performs Quasi-Orthogonal Projection (QOP) by identifying the dominant singular subspace of previously merged task vectors and projecting each new task vector onto its approximate orthogonal complement, thereby filtering out conflicting directions. Second, it applies Attention-Exclusive Fine-Tuning (AEFT), which restricts parameter updates to Transformer attention layers to promote greater orthogonality among task vectors. The authors argue that this combination allows for a better balance between stability and plasticity.

**Strengths:**

The paper is well written and presents a clear and coherent formulation of the proposed method. The motivation for addressing task interference in merging-based continual learning is relevant, especially as model merging continues to gain traction as an alternative to replay- or regularization-based CL. The authors present the mathematical formulation of their projection mechanism in detail, and the integration of an orthogonality-aware update rule with attention-exclusive fine-tuning is articulated clearly.

**Weaknesses:**

Despite its good presentation, the paper’s conceptual contribution is limited. Orthogonal updates have been extensively studied in continual learning and model merging, and the notion of constraining new updates to the orthogonal complement of previously learned subspaces has appeared in various forms (e.g., OWM, OGD, SGP, O-LoRA, and others). The proposed Quasi-Orthogonal Projection mainly rephrases this idea in the context of task-vector merging and employs a low-rank approximation via SVD, which is not a fundamentally novel insight. Similarly, restricting fine-tuning to attention layers (AEFT) has been explored previously, including in recent studies like InfLoRA [1] and O-LoRA, where adapters or low-rank components are made orthogonal across tasks, a formulation that is very close to QOMM but more computationally efficient since it updates only low-rank modules instead of full-rank weight matrices. These prior works, along with others like SD-LoRA [2] and DualPrompt [3] or CODA-Prompt [4] for continual transformers, are not discussed, even though they offer directly comparable or superior trade-offs between efficiency and orthogonality.

From a methodological standpoint, the work does not provide new theoretical insights into why quasi-orthogonal projections at the model-merging level should outperform standard orthogonal or subspace-constrained methods. The low-rank approximation and scaling factor are introduced heuristically, without analysis of stability, convergence, or computational complexity. Although the authors highlight memory efficiency, they do not quantify the computational overhead of repeated SVD operations or compare it against lighter-weight alternatives like LoRA-based orthogonalization.

In the experiments, results are limited to CL benchmarks using CLIP/ViT-B16 backbones, and while performance improvements over MagMax are consistent, they are relatively modest considering the additional computational cost. The related work section would benefit from a deeper integration of recent literature on parameter-efficient continual adaptation, especially orthogonal low-rank methods and prompt-based tuning, to better contextualize the contribution. Overall, the work feels incremental rather than offering a significant conceptual or empirical advance.

[1] Liang, Y.S. and Li, W.J., 2024. Inflora: Interference-free low-rank adaptation for continual learning. In Proceedings of the IEEE/CVF Conference on Computer Vision and Pattern Recognition (pp. 23638-23647).

[2] Wu, Y., Piao, H., Huang, L.K., Wang, R., Li, W., Pfister, H., Meng, D., Ma, K. and Wei, Y., SD-LoRA: Scalable Decoupled Low-Rank Adaptation for Class Incremental Learning. In The Thirteenth International Conference on Learning Representations.

[3] Wang, Z., Zhang, Z., Ebrahimi, S., Sun, R., Zhang, H., Lee, C.Y., Ren, X., Su, G., Perot, V., Dy, J. and Pfister, T., 2022, October. Dualprompt: Complementary prompting for rehearsal-free continual learning. In European conference on computer vision (pp. 631-648). Cham: Springer Nature Switzerland.

[4] Smith, J.S., Karlinsky, L., Gutta, V., Cascante-Bonilla, P., Kim, D., Arbelle, A., Panda, R., Feris, R. and Kira, Z., 2023. Coda-prompt: Continual decomposed attention-based prompting for rehearsal-free continual learning. In Proceedings of the IEEE/CVF conference on computer vision and pattern recognition (pp. 11909-11919).

**Questions:**

•  How does QOMM compare empirically and computationally (in terms of FLOPs, runtime, and memory footprint) with low-rank orthogonal methods such as InfLoRA or O-LoRA, which also enforce task-wise orthogonality but in a parameter-efficient manner?

•  Since AEFT restricts updates to a small subset of parameters, to what extent could the observed gains arise from the reduced update capacity rather than from increased task-vector orthogonality itself?

•  The proposed approach appears conceptually related to InfLoRA, which designs LoRA adapters that are orthogonal across tasks. Could the authors clarify in concrete terms how QOMM differs from or improves upon InfLoRA both algorithmically and computationally?

•  How does QOMM perform relative to prompt-based continual learning approaches such as DualPrompt, CODA-Prompt, or L2P, which also aim to achieve modular, interference-resistant adaptation in Transformers?

•  Finally, could the authors clearly articulate the main novelty of their contribution compared to existing orthogonal or merging-based continual learning methods, beyond combining quasi-orthogonal projection with attention-exclusive fine-tuning?

---

### Note · Authors · 2025-11-13

I have read and agree with the venue's withdrawal policy on behalf of myself and my co-authors.